# The Role of microRNA in the Regulation of Cortisol Metabolism in the Adipose Tissue in the Course of Obesity

**DOI:** 10.3390/ijms25105058

**Published:** 2024-05-07

**Authors:** Jakub Podraza, Klaudia Gutowska, Anna Lenartowicz, Michał Wąsowski, Marta Izabela Jonas, Zbigniew Bartoszewicz, Wojciech Lisik, Maurycy Jonas, Artur Binda, Paweł Jaworski, Wiesław Tarnowski, Bartłomiej Noszczyk, Monika Puzianowska-Kuźnicka, Alina Kuryłowicz

**Affiliations:** 1The Faculty of Biology and Biotechnology, Warsaw University of Life Sciences, 02-787 Warsaw, Poland; jakub.podraza@gmail.com; 2II Department of Obstetrics and Gynecology, Warsaw Medical University, 00-315 Warsaw, Poland; klaudia.gutowska@wum.edu.pl; 3LabExperts Laboratory, 93-519 Łódź, Poland; anna.lenartowicz@labexperts.com.pl; 4Department of General Medicine and Geriatric Cardiology, Medical Centre of Postgraduate Education, 00-401 Warsaw, Poland; mwasowski@cmkp.edu.pl; 5Department of Human Epigenetics, Mossakowski Medical Research Centre, Polish Academy of Sciences, 02-106 Warsaw, Poland; martajonas@imdik.pan.pl (M.I.J.); mpuzianowska@imdik.pan.pl (M.P.-K.); 6Department of Internal Medicine and Endocrinology, The Medical University of Warsaw, 02-097 Warsaw, Poland; z.bartoszewicz@wum.edu.pl; 7Department of General and Transplantation Surgery, The Medical University of Warsaw, 00-694 Warsaw, Poland; 8Department of General Surgery, Barska Hospital, 02-315 Warsaw, Poland; morjon@poczta.onet.pl; 9Department of General, Oncological and Bariatric Surgery, Medical Centre of Postgraduate Education, 00-401 Warsaw, Polandwtarnowski@cmkp.edu.pl (W.T.); 10Department of Plastic Surgery, Medical Centre of Postgraduate Education, 00-401 Warsaw, Poland; bnoszczyk@melilot.pl; 11Department of Geriatrics and Gerontology, Medical Centre of Postgraduate Education, 01-826 Warsaw, Poland

**Keywords:** cortisol metabolism, glucocorticoid receptor alpha, obesity, adipose tissue, microRNA, metabolic inflammation

## Abstract

The similarity of the clinical picture of metabolic syndrome and hypercortisolemia supports the hypothesis that obesity may be associated with impaired expression of genes related to cortisol action and metabolism in adipose tissue. The expression of genes encoding the glucocorticoid receptor alpha (*GR*), cortisol metabolizing enzymes (*HSD11B1*, *HSD11B2*, *H6PDH*), and adipokines, as well as selected microRNAs, was measured by real-time PCR in adipose tissue from 75 patients with obesity, 19 patients following metabolic surgery, and 25 normal-weight subjects. Cortisol levels were analyzed by LC-MS/MS in 30 pairs of tissues. The mRNA levels of all genes studied were significantly (*p* < 0.05) decreased in the visceral adipose tissue (VAT) of patients with obesity and normalized by weight loss. In the subcutaneous adipose tissue (SAT), *GR* and *HSD11B2* were affected by this phenomenon. Negative correlations were observed between the mRNA levels of the investigated genes and selected miRNAs (hsa-miR-142-3p, hsa-miR-561, and hsa-miR-579). However, the observed changes did not translate into differences in tissue cortisol concentrations, although levels of this hormone in the SAT of patients with obesity correlated negatively with mRNA levels for adiponectin. In conclusion, although the expression of genes related to cortisol action and metabolism in adipose tissue is altered in obesity and miRNAs may be involved in this process, these changes do not affect tissue cortisol concentrations.

## 1. Introduction

Although the gonads and adrenal glands are the main sites of steroid hormone synthesis, adipose tissue also contributes to steroidogenesis [1]. Human adipocytes express virtually all the genes encoding enzymes that are crucial for the synthesis of steroid hormones including estrogens and androgens, but also cortisol and its metabolites [1,2]. Hormones produced in adipose tissue can act in an auto- and paracrine manner, but they can also be secreted into the circulation and impact distant tissues and organs, enriching the systemic steroid pool [1,2]. Therefore, changes in adipose tissue function can have implications for the hormonal status of the whole organism. Indeed, deficiency of adipose tissue and the adipokines it produces leads to the multi-endocrine organ dysfunction observed in patients with anorexia [3]. Similarly, obesity-related changes in adipose tissue, resulting from excessive lipid accumulation, associated oxidative stress, and metabolic inflammation, also have a negative impact on the body’s endocrine balance [4]. Thus, obesity promotes, among others, the development of infertility, hyperandrogenism in women, and hypogonadism in men [4,5,6].

Cortisol is a crucial steroid hormone in the whole body homeostasis. Apart from the control of stress and inflammatory responses, cortisol is involved in the regulation of glucose, lipid, and protein metabolism, both directly and by the modulation of other hormones’ (e.g., insulin, glucagon and catecholamines) secretion [7]. Since genes encoding enzymes critical for cortisol synthesis and action are expressed in adipocytes (Figure 1), one can assume that obesity-related adipose tissue dysfunction may affect the local and systemic pool of this glucocorticoid [8].

Such a concept is supported by the fact that the clinical picture of obesity, especially abdominal obesity, resembles to some extent the phenotype of Cushing’s syndrome (a disease caused by an excess of endogenous or exogenous glucocorticoids). In fact, patients with hypercortisolism are characterized by excessive accumulation of visceral adipose tissue, the presence of hypertension, abnormal glucose metabolism, and hyperlipidemia—typical components of metabolic syndrome [13]. On the other hand, tests for the assessment of hypothalamic-pituitary-adrenal axis function in obese patients have a high degree of variability and higher cortisol levels do not translate into unfavorable metabolic profiles [14]. It is therefore worth investigating whether the action and metabolism of cortisol in adipose tissue is affected in obesity.

An additional consideration for studying obesity-related changes in adipose tissue cortisol metabolism is that glucocorticoids are key regulators of the inflammatory response. Obesity-related dysfunction of adipose tissue is manifested by infiltration of inflammatory cells and a change in secreted adipokines towards those with a pro-inflammatory profile. Inflammatory mediators exacerbate insulin resistance in adipose tissue through auto- and paracrine effects and, when released into the circulation, interfere in an endocrine manner with the function of other organs and tissues, contributing to the development of obesity-related complications [15,16].

Genes involved in cortisol metabolism and action (Figure 1) belong to the group of stress-responsive genes whose expression is modulated by environmental stimuli [17]. Recent studies have shown that microRNAs (miRNAs) play an important role in regulating gene expression in adipocytes, both in the physiological state and in adaptation to energy overload [18]. These molecules are also involved in the regulation of cortisol metabolism and action [19] and play a role in the body’s response to stress [20].

Given the phenotypic similarity between obesity and hypercortisolemia, it is plausible that excessive body weight will affect cortisol metabolism in adipose tissue. Therefore, the main aim of this study was to investigate the impact of obesity and weight loss on the expression of genes key to cortisol metabolism and action (Figure 1) in human adipose tissue. We also aimed to verify whether these changes could affect local concentrations of cortisol and selected adipokines. Then, given the role of cortisol in the regulation of carbohydrate metabolism and insulin sensitivity, we also investigated how the diagnosis of diabetes affected the expression of the genes studied. Finally, in the search for potential factors responsible for obesity-related changes in the expression of genes related to cortisol action and metabolism in adipose tissues, we investigated whether microRNA interference could play such a role.

## 2. Results

### 2.1. Impact of Obesity and Weight Loss on the Expression of Genes Involved in Cortisol Metabolism and Action in Adipose Tissue

Analysis of gene expression related to cortisol metabolism and action revealed significant differences between the tissues studied (Figure 2). As there were no differences in expression between females and males for any of the investigated genes, all analyses described were performed together on all tissues without stratification by sex.

For all genes studied, mRNA levels were lower in obese than in normal-weight subjects, and this was true for both visceral (VAT) and subcutaneous (SAT) depots. For VAT, the difference was always statistically significant (*p* < 0.05). For SAT, significant differences were found only for genes encoding glucocorticoid receptor alpha (*GR*, *p* < 0.05, Figure 2a) and 11β-hydroxysteroid dehydrogenase type 2 (*HSD11B2*, *p* < 0.0001, Figure 2d). In contrast, weight loss resulted in a significant (*p* < 0.0001) increase in the mRNA levels of each of the genes studied. In the case of 11β-hydroxysteroid dehydrogenase type 1 (*HSD11B1*, *p* < 0.0001, Figure 2c), this increase was significantly above the levels observed in the tissues of normal-weight individuals.

Obesity also had an effect on differences in the expression of genes related to cortisol metabolism and action between the depots. In normal weight-subjects, expression of *H6PDH* (*p* < 0.05, Figure 2b), *HSD11B1* (*p* < 0.0001 Figure 2c), and *HSD11B2* (*p* < 0.0001, Figure 2d) at the mRNA level was higher in VAT than in SAT and comparable in *GR* (*p* > 0.05). However, for all these genes, there was a significant decrease in expression in VAT compared to SAT in the course of obesity.

### 2.2. Diabetes Is Associated with a Significant Decrease of Expression of Genes Involved in Cortisol Metabolism and Action in Adipose Tissue

Given the important role of cortisol in the regulation of carbohydrate metabolism, including insulin sensitivity, we tested whether the mRNA levels of the above four genes differed between the tissues of obese patients diagnosed with type 2 diabetes (VAT-D, SAT-D) and normoglycemic obese subjects (VAT-ND, SAT-ND). We observed significantly lower expression levels of each gene in visceral tissues of diabetic patients than in obese subjects without diabetes: *p* < 0.001 for *GR* (Figure 3a), *p* < 0.0001 for *H6PDH* (Figure 3b), *p* < 0.01 for *HSD11B1* (Figure 3c), and *p* < 0.0001 for *HSD11B2* (Figure 3d). Surprisingly, for *HSD11B2*, its mRNA levels were significantly higher in subcutaneous tissue from diabetic patients (SAT-D) than from normoglycemic obese subjects (SAT-ND, *p* < 0.001, Figure 3d). For the other genes tested, we observed no significant differences in their expression levels between SAT-D and SAT-ND tissues.

### 2.3. Decreased Expression of Genes Involved in Cortisol Metabolism and Action in Adipose Tissues of Patients with Obesity Correlates Negatively with Levels of the Selected microRNAs

In the search for mechanisms potentially responsible for the observed differences in the expression of the studied genes in adipose tissue in the course of obesity, we investigated whether microRNAs (miRNAs) could play such a role. To this end, we used bioinformatics tools (miRTarBase, miRBase, miRDB) to select miRNAs that could potentially interact with the mRNAs of the genes studied and examined their expression in the investigated tissues (Appendix A). We then correlated the mRNA concentrations of *GR*, *H6PDH*, *HSD11B1*, and *HSD11B1* with the levels of the miRNAs studied (Figure 4, Figure 5, Figure 6 and Figure 7).

N – m We found significant negative correlations between mRNA levels of the gene encoding the glucocorticoid receptor alpha (*GR*) and levels of hsa-miR-142-3p (r_s_ = −0.4776, *p* = 0.0003), hsa-miR-561-3p (r_s_ = −0.2745, *p* = 0.0467) and hsa-miR-561-5p (r_s_ = −0.3080, *p* = 0.0249) in the visceral adipose tissues of the obese study participants (VAT-O) (Figure 4).

For the *H6PDH* gene mRNA, we observed negative correlations with hsa-miR-142-3p levels (r_s_ = −0.4787, *p* = 0.0001) and hsa-miR-579-5p levels (r_s_ = −0.3375, *p* = 0.0154), also in the VAT-O group (Figure 5).

For the mRNAs of genes encoding 11β-HSD 1 and 2 enzymes, correlations were found with the levels of selected miRNAs in both visceral (VAT) and subcutaneous (SAT) adipose tissues. For *HSD11B1*, its mRNA concentrations correlated negatively with the levels of hsa-miR-142-3p levels both in VAT-O (r_s_ = −0.3430, *p* = 0.0111) and in SAT-O (r_s_ = −0.2769, *p* = 0.0338) and with hsa-miR-561-5p levels (r_s_ = −0.3440, *p* = 0.0101) in SAT-O (Figure 6).

In turn, *HSD11B2* mRNA levels correlated negatively with the concentrations of hsa-miR-142-3p (r_s_ = −0.3667, *p* = 0.0075) and hsa-miR-579-5p (r_s_ = −0.3800, *p* = 0.0119) in VAT-O, and with levels of hsa-miR-579-3p both in VAT-O (r_s_ = −0.3589, *p* = 0.0248) and in SAT- O (r_s_ = −0.3228, *p* = 0.0269) (Figure 7).

Interestingly, a negative correlation between mRNA for *HSD11B2* and hsa-miR-579-3p and -5p was also observed in SAT-N tissues (Appendix A).

### 2.4. Impact of Cortisol Adipose Tissue Concentrations on Local Adipokine Levels

Finally, we decided to test whether the obesity-related changes in expression of the genes associated with cortisol metabolism described above translate into differences in the concentration of this hormone in adipose tissue. However, we found no statistically significant differences (*p* > 0.05) in cortisol levels between the tissues studied (Figure 8). Furthermore, we found no significant correlation between the mRNA levels of the genes studied and tissue cortisol levels (Appendix A).

Nevertheless, we decided to see if there was a correlation between local cortisol concentrations in adipose tissue and its secretory activity, namely the concentration of pro- and anti-inflammatory cytokines. These concentrations were examined at both mRNA and protein levels (Appendix A). We observed a significant correlation between the cortisol concentration in SAT-O and mRNA levels for the gene encoding adiponectin (*ADIPOQ*), Figure 9a. SAT-O cortisol levels were also negatively correlated with adiponectin protein levels, but this correlation was not statistically significant (r_s_ = −0.336, *p* > 0.05, Figure 9b).

## 3. Discussion

Endogenous glucocorticoids play a key role in maintaining homeostasis in the body, while exogenous ones have a range of clinical applications [7]. Therefore, disturbances in the expression of genes related to glucocorticoid action and metabolism can lead to serious metabolic disorders [21]. The aim of the present study was to investigate the effects of obesity on cortisol levels and the expression of genes key to its synthesis and function in adipose tissue. Given the role of cortisol in the regulation of glycemia and insulin sensitivity, we also investigated how the expression of these genes changes in the tissues of diabetic patients. Another aim of the study was to assess whether adipose tissue cortisol concentrations correlate with local inflammation severity and impact the expression of selected adipokines. Finally, attempts were also made to correlate the results of the gene expression analysis with the levels of certain microRNAs to see whether this epigenetic mechanism might be responsible for obesity-related changes in glucocorticoid metabolism and action.

In the investigated adipose tissues, the expression of the genes encoding the glucocorticoid receptor alpha and 11β-HSD1, 11β-HSD2, H6PDH enzymes was altered in obese individuals compared to normal weight-subjects without metabolic disorders. However, these differences depended on the type of gene and the adipose tissue depot.

In the case of *GR*, its expression was significantly lower (independently on the investigated depot), consistent with the results of previous studies and suggests that in the course of obesity one can expect a decreased activity of glucocorticoid receptor in adipose tissue [12,22]. However, the fact, that the highest *GR* mRNA levels were found in the SAT-PO tissues, leads to the assumption that weight loss can probably reverse the obesity-related decline in its expression. It is also worth noting that in both obese and normal-weight subjects, similar to the study of Veilleux et al., the level of *GR* expression was higher in the subcutaneous than in the visceral depot [23]. To conclude about the significance of these findings one has to analyze them in the context of the impact of obesity on the local tissue cortisol levels. Since in the presented study, there were no differences in the concentration of this glucocorticoid in the tissues obtained from obese and normal-weight subjects, one can conclude that the decreased expression of *GR* in the course of obesity can act as a protective mechanism preventing the dysfunctional adipose tissue from the additive, detrimental impact of glucocorticoids, aggravating, for example, insulin resistance or adipogenesis [24,25]. This is only speculation, but such a concept is plausible because studies in animal models show that high concentrations of glucocorticoids have a detrimental effect on adipogenesis and adipocyte function [24,25].

While decreased expression of *GR* in individuals with obesity has been described previously [12,22], our study was the first to report that grade III obesity in humans is associated with a decreased concentration of mRNA levels for *HSD11B1* (encoding an enzyme responsible for cortisone to cortisol conversion) in visceral adipose tissue. In the studies by Michailidou et al. and by Veilleux et al., 11β-HSD1 levels and activity in VAT were positively correlated with the body mass index (BMI) and visceral adiposity [22,23]. When considering the possible reasons for this discrepancy, the different characteristics of the study participants can be pointed out: whereas the tissues examined in our study were obtained from patients with grade III obesity undergoing bariatric surgery (mean BMI = 46.27 kg/m^2^), previous reports included participants with grade I obesity or overweight (mean BMI 32.7 kg/m^2^ and 27.1 kg/m^2^, respectively) [22,23]. In addition, another study showed that in some cases, the expression of *HSD11B1* is lower in the VAT obtained from patients with obesity than from normal-weight patients, and that there is no regularity in this phenomenon [26]. Nevertheless, similarly to previous researchers, we observed an impact of obesity on the *HSD11B1* expression between the visceral and subcutaneous depots: it was significantly higher in the VAT of normal-weight subjects, while in the course of obesity, its mRNA levels were higher in the SAT (Figure 2) [27].

To have a better picture of the impact of obesity on the cortisol metabolism in the adipose tissue, we also analyzed the expression of the *H6PDH*. As in the case of the other investigated genes, its mRNA levels were significantly decreased in the visceral adipose tissue of the patients with obesity, which is consistent with the findings of Veilleux et al. [23]. The hexose-6-phosphate glucose 1-dehydrogenase causes the conversion of NADP to NADPH, which can be used by 11β-HSD1 in the reaction of converting cortisone to active cortisol [10]. Previous studies have shown that the expression of these two genes and the activity of the enzymes they encode are positively correlated [28]. In our study, the expression profile of these two genes was very similar in the tissues examined.

Given the role of glucocorticoids in the regulation of glucose metabolism, we decided to analyze whether the diabetic subjects differed in any way from the rest of the study population in terms of the expression of the investigated genes. When we compared the subgroup of patients with diabetes and obesity to normal-weight subjects in terms of the expression of the genes studied, we found the same trends as the entire study cohort of obese patients. However, when a group of obese diabetics was compared with the normoglycemic obese patients, expression of all studied genes in the diabetic group was lower than in the case of normoglycemic people with obesity, and this difference was particularly pronounced in VAT. Assessing whether lower expression of genes related to cortisol action and metabolism in adipose tissues of patients with type 2 diabetes may have clinical relevance requires verification by functional studies. Past studies suggested that response to glucocorticoids in type 2 diabetes can manifest as resistance or hyperactivity in a tissue-specific manner (reviewed in [29]).

What is the most intriguing is the fact that the obesity-associated changes in the expression of 11β-hydroxysteroid dehydrogenase in adipose tissue did not transfer to the altered tissue cortisol level. In order to obtain reliable results, these determinations were carried out using the LC-MS/MS reference method, as opposed to the radioimmunoassay previously used [23,30]. Our observation that excessive adiposity-related changes in the expression of enzymes responsible for cortisol metabolism does not translate into differences in adipose tissue concentrations of this hormone is consistent with clinical data—biochemically verified hypercortisolemia is rarely found in patients with obesity, who more likely have functional hypercortisolemia [14,17,31,32]. Although the tissues were collected in the forenoon, it cannot be excluded that the lack of significant differences in cortisol concentrations between the study groups is due to a lack of complete time synchronization of tissue collection.

Nevertheless, we tested whether tissue cortisol concentrations could influence local inflammatory and secretory activity. To this end, we correlated them with expression studies of cytokines and adipokines produced in adipose tissue. We found that cortisol concentrations in the subcutaneous adipose tissues of obese patients (SAT-O) correlated negatively with *ADIPOQ* mRNA levels. The effect of glucocorticoids on the expression of the gene encoding adiponectin has been extensively studied in vitro, in vivo, and in the clinical setting, with conflicting results [33]. As our finding of a correlation between tissue cortisol levels and ADIPOQ expression is not supported by functional studies, and we have not evaluated expression of any inducible GR target, we cannot speculate about the activity of this receptor in the adipose tissue of obese individuals. It can be only assumed that the mechanisms responsible for obesity-related changes in adiponectin secretion are highly complex.

This work also attempted to elucidate the possible mechanisms responsible for the observed differences in gene expression key to cortisol metabolism and action in the course of obesity. Research in recent years has drawn attention to the role of miRNAs as important regulators of gene expression in adipose tissue [18]. Therefore, we investigated the impact of obesity and weight loss on the local expression of the selected miRNAs which, based on in silico analysis and/or functional evidence, may play a role in the regulation of gene expression related to cortisol metabolism and action. Of the miRNAs tested, the most promising results were obtained for hsa-miR-142-3p, whose concentrations in VAT-O correlated negatively with mRNA levels for all studied genes. Given the role of miR-142-3p in regulating stress-induced expression of the gene encoding GRα in the central nervous system of rats fed a high-fat diet, we believe that this result should be verified in functional studies [34].

Recent studies have also implicated hsa-miR-561 and hsa-miR-579 in the regulation of gene expression related to cortisol metabolism in human hepatocytes [35]. Our research suggests that these findings are also applicable in the case of adipose tissue. Based on correlation analyses, hsa-miR-561-5p appears to be a negative regulator of *11BHSD1* expression, whereas hsa-miR-579-3p, hsa-miR-579-3p, and hsa-miR-579-5p are negative regulators of *11BHSD2* expression in adipose tissues of obese patients. The latter miRNA may also be involved in the obesity-related downregulation of *H6PDH* in adipose tissue.

Although our results drawn from a large group of participants may represent a new contribution to the understanding of changes in cortisol metabolism in obesity, they have several undoubted limitations. First, the study design used renders the results purely descriptive. Next, gene expression studies were performed at the mRNA level, which does not always translate to protein concentration. Moreover, correlation analyses on miRNAs need to be verified in functional studies. In addition, tissue cortisol concentrations were not measured in all tissues examined, but only in a representative group of 30 individuals. Another potential limitation of our work is that the studied cohort was predominantly female (which is typical for patients undergoing bariatric surgery) and local differences in sex hormone levels could affect the results. However, in our cohort, we found no significant differences in the expression profile of the genes studied between the sexes.

## 4. Materials and Methods

### 4.1. Study Participants

Gene and miRNA expression analyses were performed on adipose tissue samples obtained from 75 individuals with obesity (62 females and 13 males). The mean age of the subjects was 41.43 years (SD = 10.24, range 20–62 years). Based on the WHO classification, all patients were diagnosed with class III obesity: mean body mass index (BMI) was 46.27 kg/m2 (SD = 5.51, range: 35.43–59.52 kg/m^2^). The mean waist circumference in the study group was 1.24 m (SD = 0.18 m; range: 0.97–1.67 m), while the mean body fat percentage was 47.09% (SD = 5.16%; range: 32.64–59.52%). Twenty-five individuals (33.33%) from the study group were also diagnosed with type 2 diabetes or pre-diabetes (impaired glucose tolerance, abnormal fasting glucose). Forty-two patients (56%) were diagnosed with hypertension and 46 (61.33%) with hyperlipidemia. Using the criteria proposed by the International Diabetes Federation for the European population (waist circumference greater than 80 cm in women and 94 cm in men, and the presence of at least 2 of 3 disorders: dyslipidemia, hypertension, carbohydrate intolerance), 40 patients (53.33%) were diagnosed with metabolic syndrome.

The control group consisted of 25 individuals (20 women, 5 men) aged 23 to 62 years (mean age was 47.71 years, SD = 13.53) whose BMI was within the normal range (20.1–24.93 kg/m^2^, mean 23.28 kg/m^2^, SD = 1.65 kg/m^2^). Although body composition was not assessed in the control group, based on normal results of basic blood tests and a negative history of chronic disease (including components of the metabolic syndrome), all individuals in the control group were considered to be metabolically healthy. Clinical characteristics of the study groups can be found in Appendix A.

Cortisol concentrations were measured in 30 pairs of visceral and subcutaneous adipose tissue from 22 obese patients and 8 normal-weight subjects. The expression levels of the genes and miRNAs studied in the subgroup in which cortisol determinations were performed were not significantly different from those observed in the entire study cohort.

### 4.2. Tissues

Visceral (VAT) and subcutaneous (SAT) adipose tissue samples were obtained from obese patients undergoing bariatric surgery. SAT samples were taken from the lower abdomen during abdominoplasty 24 months after surgery from 19 patients in the study group who had successfully lost weight through bariatric surgery. The abdominal cavity is not opened during abdominoplasty. Therefore, it was not possible to collect VAT samples from these patients. The control adipose tissue samples were collected from normal weight patients during elective cholecystectomy (VAT) or inguinal hernia surgery (SAT). Immediately after collection, all tissues were snap frozen in liquid nitrogen and stored at −80 °C until testing. The study was carried out according to the guidelines of the declaration of Helsinki. All procedures included in the adipose tissue collection protocol were approved by the Bioethics Committee of the Medical University of Warsaw (decision no. KB 147/2009 issued on 28 July 2009, KB 91/A/2010 issued on 19 July 2010, KB 117/A/2011 issued on 14 November 2011, and KB 38/A/2022 issued on 16 May 2022). Written informed consent was obtained from all study participants.

### 4.3. Genes and miRNAs Expression Analysis

Isolation of total RNA from adipose tissue, reverse transcription (separately for mRNA and miRNA), and real-time PCR were performed as described previously [36]. Primers used to analyze the expression of genes related to glucocorticoid action and metabolism are listed below.

β-actin (*ACTB*—reference gene):

Forward Primer: CAGCCTGGATAGCAACGTAC

Reverse Primer: TTCTACAATGAGCTGCGTGTG

Glucocorticoid receptor alpha (*GR*):

Forward Primer: CCTGTCTGTACCTAACGCCCTAT

Reverse Primer: GGGTGTCTAGCCATTTTTGCCATATT

Hexose-6-phosphate dehydrogenase (*H6PDH*):

Forward Primer: TGCTTTTACCTCTCGTCCACTG

Reverse Primer: GTGGGTTTTTGTTTTATAGGGAGACT

11β-hydroxysteroid dehydrogenase type 1 (*HSD11B1*):

Forward Primer: TCATTCTCAACCACATCACCAACAC

Reverse Primer: CCAGCCAGAGAGGAGACGACAAC

11β-hydroxysteroid dehydrogenase type 1 (*HSD11B2*):

Forward Primer: TCAGGCTGTGACTCTGGTTTTGGCA

Reverse Primer: CGGGGCTGTTCAACTCCAATAC

Adiponectin (*ADIPOQ*)

Forward Primer: GGTCTCGAACTCCTGGCCTA

Reverse Primer: TGAGATATCGACTGGGCATGGT

Resistin (*RETN*)

Forward Primer: GCTGTTGGTGTCTAGCAAGAC

Reverse Primer: CAGCCTGGATAGCAACGTACA

Interleukin 1β (*IL1B*)

Forward Primer: CACCAAGCTTTTTTGCTGTGAGT

Reverse Primer: GCACGATGCACCTGTACGAT

Interleukin 6 (*IL6*)

Forward Primer: CCTTCGGTCCAGTTGCCTTC

Reverse Primer: GTGGGGCGGCTACATCTTTG

Interleukin 8 (*IL8*)

Forward Primer: CACCGGAAGAACCATCTCACT

Reverse Primer: TCAGCCCTCTTCAAAAACTTCTCC

Interleukin 15 (*IL15*)

Forward Primer: GGATTTACCGTGGCTTTGAGTAATGAG

Reverse Primer: GAATCAATTGCAATCAAGAAGTG

For microRNA analysis, available literature sources were used to evaluate and select microRNAs that may directly or indirectly correlate with glucocorticoid gene receptor expression. Next, the miRprimer2 program dedicated to the design of miRNA primers was used. The designed primers (listed below) were verified in the miRanda software.

hsa-miR-103a-3p reference microRNA:

Forward Primer: GCAGAGCAGCATTGTACAG

Reverse Primer: GGTCCAGTTTTTTTTTTTTTTTCATAG

hsa-miR-561-3p:

Forward Primer: GCAGCAAAGTTTAAGATCCTTG

Reverse Primer: GGTCCAGTTTTTTTTTTTTTTTACTTC

hsa-miR-561-5p:

Forward Primer: CGCAGATCAAGGATCTAAACTT

Reverse Primer: TCCAGTTTTTTTTTTTTTTTGGCA

hsa-miR-579-3p:

Forward Primer: CGCAGTTCATTTGGTATAAACC

Reverse Primer: GGTCCAGTTTTTTTTTTTTTTTAATCG

hsa-miR-579-5p:

Forward Primer: GCGGTTTGTGCCAGATG

Reverse Primer: GGTCCAGTTTTTTTTTTTTTTTCGT

hsa-miR-142-3p:

Forward Primer: CGCAGTGTAGTGTTTCCT

Reverse Primer: GGTCCAGTTTTTTTTTTTTTTTCCA

### 4.4. Isolation of a Protein Fraction from Adipose Tissue and Measurement of Cytokine Concentrations

A previously described method was used to isolate the adipose tissue protein fraction [37,38]. Interleukin (IL) 1β, 6, 8, 15, adiponectin, and resistin concentrations in adipose tissue protein extracts were measured using an ELISA-based chemiluminescent Q-plex Custom array (Quansys Bioscience, West Logan, UT, USA). A Molecular Imager Versa Doc™ MP 5000 system (Bio-Rad, Hercules, CA, USA) was used to assess luminescence according to the manufacturer’s guidelines. Results were analyzed using Q-View software version 2.17 (Quansys Bioscience, West Logan, UT, USA). Measurements of adipokine concentrations in adipose tissue were normalized to total protein concentrations in protein extracts. Mean total protein concentrations in VAT and SAT extracts from obese and normal-weight participants were not significantly different (*p* > 0.05).

### 4.5. Measurement of Tissue Cortisol Concentrations

Two hundred and fifty mg of homogenized adipose tissue was mixed with 1.2 mL of a mixture of ethanol and ethyl acetate (1:1, *v*/*v*) with an internal standard (cortisol-D4, Supelco Bellefonte, PA, USA). Samples were extracted by shaking for 10 min and sonication for 5 min at room temperature, followed by centrifugation (10,000× *g*, 5 min, 4 °C). After centrifugation, 1 mL of the upper layer was collected, transferred to a glass chromatography vial and evaporated under a nitrogen stream (60 °C, 50 min). The residue after evaporation was reconstituted in 360 µL of methanol, and after 2 min of shaking and sonication, 1440 µL of water was added and the samples were vortexed for 30 s. The samples were then purified by solid phase extraction (SPE). Prior to loading, Sep-Pak C18 SPE columns (2 g; Waters, Milford, MA, USA) were conditioned with 12 mL (2 × 6 mL) methanol followed by 12 mL (2 × 6 mL) water. The adipose tissue extracts (1.8 mL) were then loaded onto SPE columns. In the next step, the columns were washed with 12 mL (2 × 6 mL) of water followed by 12 mL (2 × 6 mL) of aqueous 5% (*v*/*v*) methanol. The analytes were eluted with 9 mL (3 × 3 mL) of methanol into clean glass vials. The eluates were evaporated under a stream of nitrogen (60 °C, 3.5 h). Finally, the samples were resuspended in 400 µL of methanol and, after 2 min of shaking and sonication, 600 µL of water was added and the samples vortexed for 30 s. The samples were then subjected to LC-MS/MS analysis.

### 4.6. Statistical Analysis

Normality of distribution and homogeneity of variance for selected parameters were checked using Shapiro–Wilk and Levene’s tests. Differences in mRNA and microRNA expression levels and tissue adipokine concentrations were analyzed using one-way ANOVA. For multiple comparison correction, the Tukey’s multiple comparison test was used. Correlations between the studied parameters were analyzed using Spearman’s correlation test. A value of *p* < 0.05 was used as the threshold for statistical significance (*p* is defined as the coefficient of statistical significance). All statistical analyses were performed using Statistica v.10 (StatSoft, Tulsa, OK, USA) and GraphPad Prism v.7 (GraphPad Software, San Diego, CA, USA).

## 5. Conclusions

The concept that human obesity is associated with impaired cortisol metabolism is supported by the similar clinical picture of metabolic syndrome and hypercortisolemia. In this study, we observed that mRNA concentrations of genes key to cortisol action and metabolism are altered in adipose tissues of obese patients and that weight loss normalizes their expression. Negative correlations between mRNA levels for genes encoding GR and 11β-HSD1, 11β-HSD2, H6PDH enzymes, and selected miRNAs suggest that this epigenetic mechanism may be responsible for obesity-related differences in their expression. However, the observed changes did not translate into differences in cortisol concentrations in the tissues studied, although this hormone may locally regulate the secretory activity of adipocytes.

## Figures and Tables

**Figure 1 ijms-25-05058-f001:**
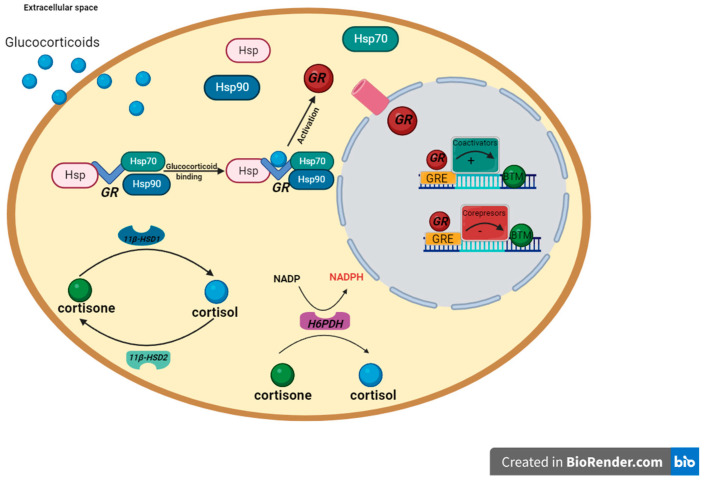
The simplified pathway of glucocorticoid receptor signaling and the action of enzymes involved in glucocorticoid metabolism in adipocytes, based on [9,10,11]. In the absence of glucocorticoid, the glucocorticoid receptor (GR) is located in the cytoplasm and forms a complex with chaperones (hsp90, p23, hsp70) as well as immunophillins (FKBP51, FKBP52). Thanks to the actions of these proteins, the GR is maintained in a transcriptionally inactive conformation, which in turn allows the binding of a ligand with high affinity to this receptor. The glucocorticoid receptor alpha (GRα) is the only active receptor for glucocorticoids in human adipose tissue [12]. Physiologically, cortisol is the most common glucocorticoid in humans. In its free, non-globulin-bound form, cortisol crosses the plasma membrane. The availability of this form of cortisol in the cell is controlled by enzymes that are antagonists to each other. 11β-hydroxysteroid dehydrogenase type 1 (11β-HSD1) is responsible for conversion of inactive cortisone to an active cortisol. In contrast, 11β-hydroxysteroid dehydrogenase type 2 (11β-HSD2) catalyzes the opposite reaction, i.e., oxidizes active cortisol into its inactive form [9]. If hexose-6-phosphate dehydrogenase (H6PDH) is present, the equilibrium can favor the activity of 11β-HSD1. H6PDH regenerates NADPH, which increases the activity of 11β-HSD1, and decreases the activity of 11β-HSD2. Once in the cell nucleus, GR binds to its responsive elements (GRE) and thus regulates the expression of target genes.

**Figure 2 ijms-25-05058-f002:**
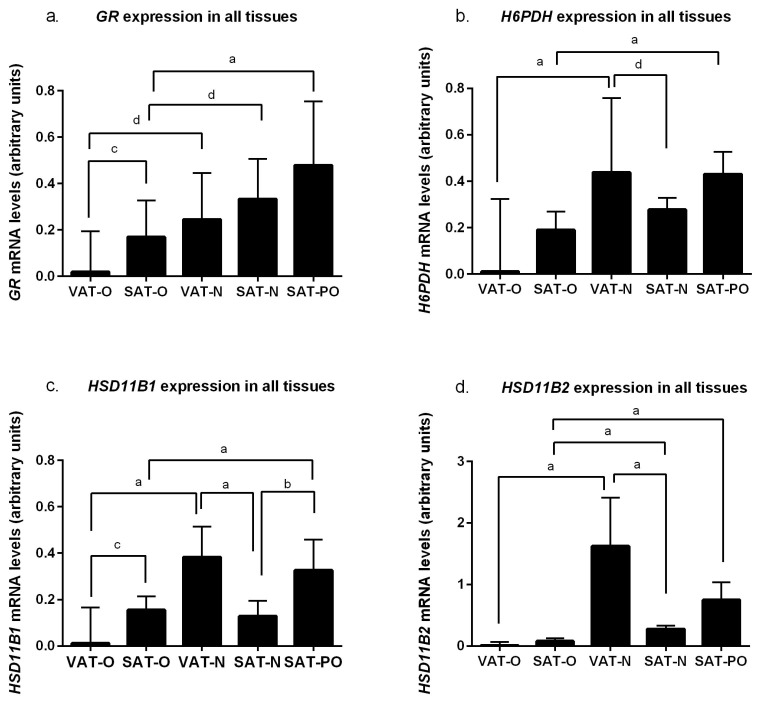
mRNA levels of genes encoding glucocorticoid receptor alpha (*GR*), (**a**); hexose-6-phosphate dehydrogenase (*H6PDH*), (**b**); 11β-hydroxysteroid dehydrogenase type 1 (*HSD11B1*), (**c**); and 11β-hydroxysteroid dehydrogenase type 2 (*HSD11B2*), (**d**) in the visceral (VAT) and subcutaneous (SAT) adipose tissues of the obese individuals, before (O) and after surgically induced weight loss (PO), and in the normal-weight subjects (N). Results normalized to β-actin (*ACTB*) mRNA concentration are presented as the median with the interquartile range. “a” *p* < 0.0001; “b” *p* < 0.001; “c” *p* < 0.01; “d” *p* < 0.05.

**Figure 3 ijms-25-05058-f003:**
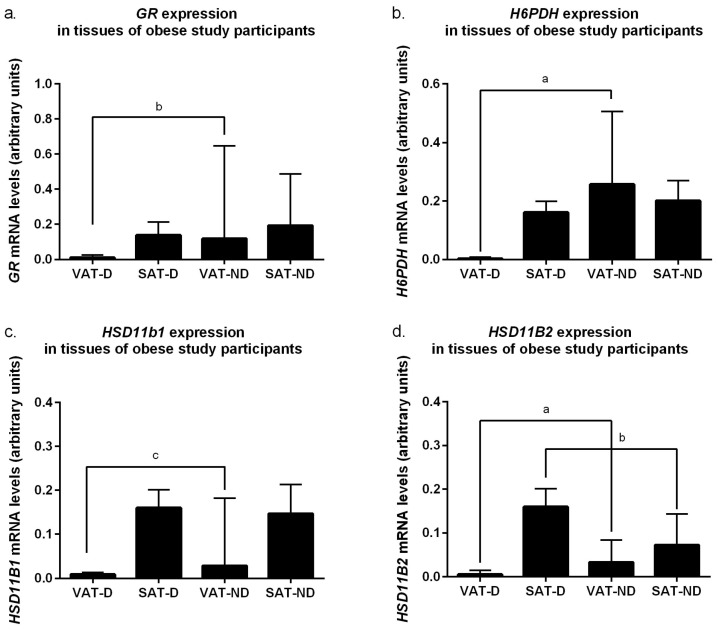
mRNA levels of genes encoding glucocorticoid receptor alpha (*GR*), (**a**); hexose-6-phosphate dehydrogenase (*H6PDH*), (**b**); 11β-hydroxysteroid dehydrogenase type 1 (*HSD11B1*), (**c**); and 11β-hydroxysteroid dehydrogenase type 2 (*HSD11B2*), (**d**) in the visceral (VAT) and subcutaneous (SAT) adipose tissues of obese individuals (O) diagnosed with type 2 diabetes (D) and without diabetes (ND). Results normalized to β-actin (*ACTB*) mRNA concentration are presented as the median with the interquartile range. “a” *p* < 0.0001; “b” *p* < 0.001; “c” *p* < 0.01.

**Figure 4 ijms-25-05058-f004:**
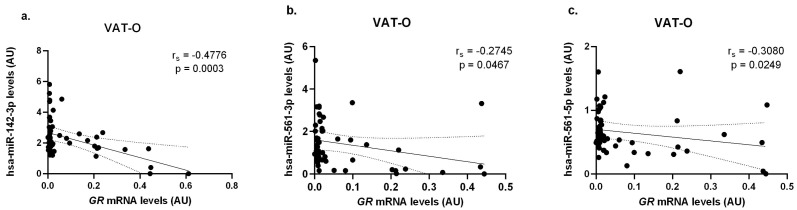
Correlations between *GR* mRNA concentrations and hsa-miR-142-3p (**a**), hsa-miR-561-3p (**b**), and hsa-miR-561-5p (**c**) in the visceral adipose tissue of obese study participants (VAT-O). Black dots represent particular participants. Lines represent linear regression analysis with 95% CI interval.

**Figure 5 ijms-25-05058-f005:**
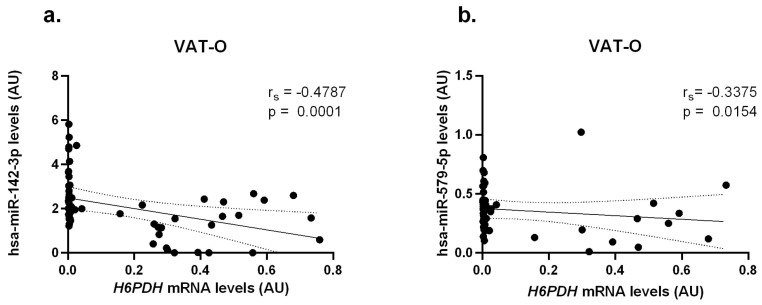
Correlations between *H6PDH* mRNA concentrations and hsa-miR-142-3p (**a**) and hsa-miR-579-5p (**b**) in the visceral adipose tissue of obese study participants (VAT-O). Black dots represent particular participants. Lines represent linear regression analysis with 95% CI interval.

**Figure 6 ijms-25-05058-f006:**
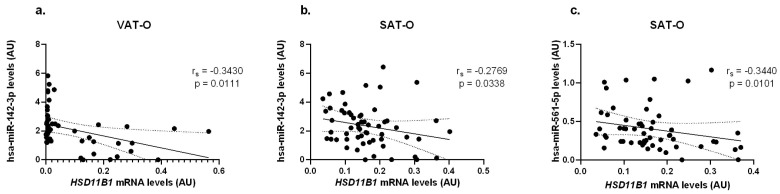
Correlations between *HSD11B1* mRNA concentrations and hsa-miR-142-3p (**a**,**b**) and hsa-miR-561-5p (**c**) levels in the visceral (VAT) and subcutaneous (SAT) adipose tissues of obese study participants (O). Black dots represent particular participants. Lines represent linear regression analysis with 95% CI interval.

**Figure 7 ijms-25-05058-f007:**
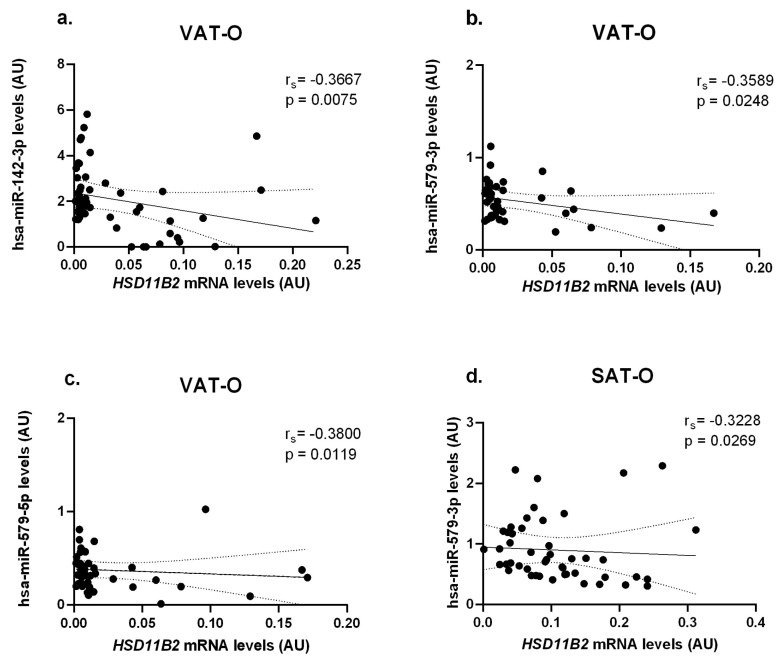
Correlations of *HSD11B2* mRNA concentrations with hsa-miR-142-3p (**a**), hsa-miR-579-3p (**b**), and hsa-miR-579-5p (**c**) levels in the visceral (VAT) and with hsa-miR-579-3p (**d**) in the subcutaneous (SAT) adipose tissues of obese study participants (O). Black dots represent particular participants. Lines represent linear regression analysis with 95% CI interval.

**Figure 8 ijms-25-05058-f008:**
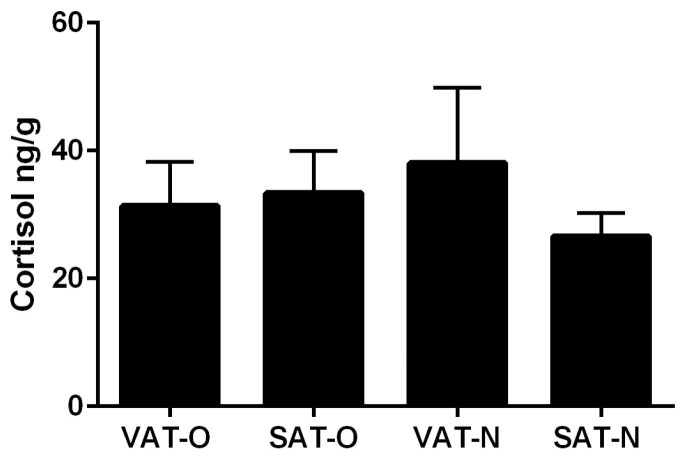
Cortisol levels in the visceral (VAT) and subcutaneous (SAT) adipose tissues of obese individuals (O) and in normal-weight subjects (N). Results are presented as the median with the interquartile range.

**Figure 9 ijms-25-05058-f009:**
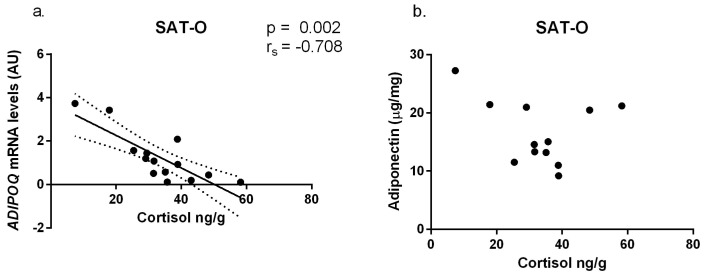
Correlations of *ADIPOQ* mRNA (**a**) and adiponectin (**b**) concentrations with cortisol levels in subcutaneous adipose tissues of obese study participants (SAT-O). Black dots represent particular participants. Lines represent linear regression analysis with 95% CI interval. Adiponectin protein concentrations (µg) are normalized to the tissue protein content (mg).

## Data Availability

The data presented in this study are available on request from the corresponding author.

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
