# Peer review of "The Role of microRNA in the Regulation of Cortisol Metabolism in the Adipose Tissue in the Course of Obesity"

_ijms, 2024, doi:10.3390/ijms25105058_

Round 1

Reviewer 1 Report

Comments and Suggestions for Authors

Summary: In this manuscript the authors sought to examine the association between obesity, diabetes, or weight loss and the mRNA expression of key genes involved in cortisol metabolism in both visceral and subcutaneous adipose tissues. Further, they aimed to link expression of these mRNAs to targeted micro-RNAs, tissue cortisol concentrations and tissue adipokine concentrations. The authors use a large sample set of adipose tissues to show that mRNA expression of key genes involved in glucocorticoid action are decreased in both in people with obesity and diabetes across adipose depots, and that weight loss results in higher expression of these genes. Further, the authors show that expression of specific micro-RNAs negatively correlate with target genes in visceral and subcutaneous adipose tissues of obese individuals. Finally they show, despite differences in mRNA expression of genes involved in metabolism of cortisol are differentially expressed between normal weight individuals and those with obesity, there was no detectable difference in tissue cortisol concentrations.

General Comments:

The manuscript is generally well written and well organized. The authors do well to connect different results in a logical manner. The large number biopsies for this study is a major strength, as well as data from participants with varying metabolic conditions. The correlations with micro-RNAs to target genes is interesting and closely reflects what the authors found in a previous study using similar methods targeting the estrogen receptor (doi: 10.3390/ijms23115989.).

Methods: The laboratory based methods are appropriate for addressing this study’s goals, and are generally well described, some statistical analyses may need to be redone (see specific comments).  Given this study uses a sample set previously examined by this group, it would be helpful either to include a subject characteristics table or cite the previous paper (already cited for mRNA analysis). A significant weakness for this study is that the authors do not mention the time of biopsy for the different groups. Given the important role of circadian variation in circulating cortisol, and the potential variability in the time of specific procedures, this information should be included, or at least mentioned in the limitations of the study. There is no de-facto measure of glucocorticoid action in the study, the authors should consider running a primary inducible target of the GR, such as GILZ or others (doi: 10.1073/pnas.0407008101.). Statistical tests do not account for multiple comparisons, especially in figures 2 and 3. It is recommended to use ANOVAs with appropriate post-tests to correct. No statement indicating these studies were carried out according to the guidelines of the declaration of Helsinki is included.

Results: The results are well described, and graphs are clear generally. With the above suggestion of ANOVAS, differences between groups may be better represented using letter codes. While it is understood that the density of correlations is already high, the lack of similar graphs, or mention of data, for normal weight individuals is of concern, especially if the intention is to show that obesity is linked to the expression of these microRNAs and that they, in part, explain the differences in expression of genes related to cortisol metabolism. Further if normal weight participants show similar trends, it lends weight to the hypothesis that the microRNAs are important in the expression of these genes. If miRNAs were not measured in this normal weight group, it is not clear from the methods that they were only used for non-miRNA studies. Regardless of significance, data for adiponectin protein should be shown in figure 9 (line 222).

Discussion: While comprehensive, aspects of the discussion feel somewhat overstated in terms of impact- see specific comments below.

252-258- It is unclear whether this statement is referring to a potential therapeutic approach of decreasing GR or whether it’s an intrinsic defense mechanism to limit metabolic complications. If the latter, the study does not support this. Metabolic improvement in adipocyte specific GR knockouts is not a validation that suppression of the GR is a defense mechanism in the course of obesity. Please clarify.

267-268- The authors state that differences in sex distribution may explain difference between their study and the Michailidou/Veilleux studies. The authors have the ability to do a sex stratified analysis, or analysis without males, especially given their significant proportions of females in their population; to see if this rectifies the difference, and should be included if they are claiming this as an explanation.

273-4- A reference to figure 2 would be helpful here.

276-290- They have data refuting these conjectures of relative activity, given the lack of difference of active cortisol among all tissues (figure 8), recommend removing this paragraph or including reference to this data with a recontextualization of the differences.

297-298- The observations reported in this study are not consistent with this description. There is no correlation shown between the levels of 11BHSD1, H6PDH and concentrations of Cortisol (the only potential measure of 11BHSD1 activity reported in this study)

308-311- The associative data presented in this study cannot provide direction of these effects, it is possible that disruptions in Glucocorticoid activity could lead to changes consistent with type 2 diabetes. Consider language that conveys this uncertainty of direction.

311-313 It is unclear what is meant by this sentence. Is it hat this previous review implied that metabolic perturbations in type 2 diabetes may lead to disruptions in GC action? Please clarify.

Comments on the Quality of English Language

I have very minor concerns on the quality of English Language

Author Response

Reviewer 1

1) The manuscript is generally well written and well organized. The authors do well to connect different results in a logical manner. The large number biopsies for this study is a major strength, as well as data from participants with varying metabolic conditions. The correlations with micro-RNAs to target genes is interesting and closely reflects what the authors found in a previous study using similar methods targeting the estrogen receptor (doi: 10.3390/ijms23115989.)

We would like to thank the reviewer for the positive reception of our work and the comments, which will undoubtedly improve the quality of the manuscript.

2) Methods: The laboratory based methods are appropriate for addressing this study’s goals, and are generally well described, some statistical analyses may need to be redone (see specific comments). Given this study uses a sample set previously examined by this group, it would be helpful either to include a subject characteristics table or cite the previous paper (already cited for mRNA analysis).

We would like to thank the Reviewer for this valid comment - in the revised version of the manuscript we have both added an appropriate reference to the paper where the study group was described and prepared a Supplementary Table S2 with clinical data of the study participants.

“Clinical characteristics of the study groups can be found in Supplementary Table S1.” (Page 11, lines 386-387)

3) A significant weakness for this study is that the authors do not mention the time of biopsy for the different groups. Given the important role of circadian variation in circulating cortisol, and the potential variability in the time of specific procedures, this information should be included, or at least mentioned in the limitations of the study

We agree with the reviewer that the lack of synchronization in tissue collection can be a serious source of bias in our measurements and, as suggested, we have included this fact in the discussion. We analyzed the medical data of the patients included in the study and verified that the procedures were performed in the morning - i.e. between 8 am and 11.00 am, a time when physiological serum cortisol concentrations are rather high. Another issue is whether cortisol concentrations in adipose tissue are also subject to circadian rhythm - as this is dependent on the action of the hypothalamic-pituitary-adrenal axis. However, it cannot be ruled out that the local activity of enzymes involved in the metabolism of this hormone plays a predominant role in the regulation of tissue cortisol concentrations. Verification of this hypothesis would require the determination of cortisol concentrations in adipose tissue samples taken from the same individual at different times of the day. While this would be theoretically possible for subcutaneous tissues, it is technically and ethically impossible for visceral deposit. These considerations are included in the discussion in the revised version of the manuscript.

“Although the tissues were collected in the forenoon, it cannot be excluded that the lack of significant differences in cortisol concentrations between the study groups is due to a lack of complete time synchronisation of tissue collection.” (Page 10, lines 319-322)

4) There is no de-facto measure of glucocorticoid action in the study, the authors should consider running a primary inducible target of the GR, such as GILZ or others (doi: 10.1073/pnas.0407008101.).

We agree with the reviewer that studying the expression of an inducible GR target, such as GILZ, would allow us to infer the activity of these receptor signalling pathways in adipose tissue during obesity and weight loss. Although the expression of the adipokines analysed in this paper is partly regulated by cortisol, it is rather indirect. We have mentioned this limitation in the revised version of the manuscript.

“As our finding of a correlation between tissue cortisol levels and ADIPOQ expression is not supported by functional studies, and we have not evaluated expression of any inducible GR target, we cannot speculate about the activity of this receptor in the adipose tissue of obese individuals. It can be only assumed that the mechanisms responsible for obesity-related changes in adiponectin secretion are highly complex.”(Page 10, lines 329-334)

5) Statistical tests do not account for multiple comparisons, especially in figures 2 and 3. It is recommended to use ANOVAs with appropriate post-tests to correct.

As suggested by the reviewer, in the revised version of the manuscript we have added information on the results of one-way ANOVA tests with a correction for multiple comparisons. Accordingly, we have modified the “Results” and "Statistical Analyses" sections as well as supplementary data.

“For all genes studied, their mRNA levels were lower in obese than in normal weight subjects, and this was true for both visceral (VAT) and subcutaneous (SAT) depots. For VAT, the difference was always statistically significant (p < 0.05). For SAT, significant differences were found only for genes encoding glucocorticoid receptor alpha (GR, p < 0.05, Figure 2a) and 11β-hydroxysteroid dehydrogenase type 2 (HSD11B2, p < 0.0001, Figure 2d). In contrast, weight loss resulted in a significant (p < 0.0001) increase in the mRNA levels of each of the genes studied. In the case of 11β-hydroxysteroid dehydrogenase type 1 (HSD11B1, p < 0.0001, Figure 2c) this increase was significantly above the levels observed in the tissues of normal-weight individuals.

Obesity also had an effect on differences in the expression of genes related to cortisol metabolism and action between the depots. In normal weight subjects, expression of H6PDH (p < 0.05, Figure 2b), HSD11B1 (p < 0.0001 Figure 2c) and HSD11B2 (p < 0.0001, Figure 2d) at the mRNA level was higher in VAT than in SAT and comparable in GR (p > 0.05).” (Pages 3-4, lines 126-139)

“We observed significantly lower expression levels of each gene in visceral tissues of diabetic patients than in obese subjects without diabetes: p < 0.001 for GR (Figure 3a), p < 0.0001 for H6PDH (Figure 3b), p < 0.01 for HSD11B1 (Figure 3c) and p < 0.0001 for HSD11B2 (Figure 3d). Surprisingly, for HSD11B2, its mRNA levels were significantly higher in subcutaneous tissue from diabetic patients (SAT-D) than from normoglycemic obese subjects (SAT-ND, p < 0.001, Figure 3d).” (Pages 4-5, lines 153-159)

“Differences in mRNA and microRNA expression levels and tissue adipokine concentrations were analyzed using one-way ANOVA. For multiple comparison correction, the Tukey’s multiple comparison test was used.”(Page 14, lines 500-503)

6) No statement indicating these studies were carried out according to the guidelines of the declaration of Helsinki is included.

We thank the reviewer for this valid comment; information about the study protocol's compliance with the Helsinki Declaration has been included in the revised version of the manuscript.

“The study was carried out according to the guidelines of the declaration of Helsinki.” (Page 12, line 401-402)

Results:

7) The results are well described, and graphs are clear generally. With the above suggestion of ANOVAS, differences between groups may be better represented using letter codes.

We thank the reviewer for this good point - in the revised version of the manuscript, we have used letter codes to facilitate the interpretation of the results. The same changes have been made to supplementary figures S 1-3.

Figure 2. mRNA levels of genes encoding glucocorticoid receptor alpha (GR, a), hexose-6-phosphate dehydrogenase (H6PDH, b), 11β-hydroxysteroid dehydrogenase type 1 (HSD11B1, c) and 11β-hydroxysteroid dehydrogenase type 2 (HSD11B2, d) in the visceral (VAT) and subcutaneous (SAT) adipose tissues of the obese individuals before (O) and after surgically induced weight loss (PO) and in the normal-weight subjects (N). Results normalized to β-actin (ACTB) mRNA concentration are presented as the median with the interquartile range. “a” p < 0.0001; “b” p < 0.001; “c” p < 0.01; “d” p < 0.05.

Figure 3. mRNA levels of genes encoding glucocorticoid receptor alpha (GR, a), hexose-6-phosphate dehydrogenase (H6PDH, b), 11β-hydroxysteroid dehydrogenase type 1 (HSD11B1, c) and 11β-hydroxysteroid dehydrogenase type 2 (HSD11B2, d) in the visceral (VAT) and subcutaneous (SAT) adipose tissues of the obese individuals (O) diagnosed with type 2 diabetes (D) and without diabetes (ND). Results normalized to β-actin (ACTB) mRNA concentration are presented as the median with the interquartile range. “a” p < 0.0001; “b” p < 0.001; “c” p < 0.01.

8) While it is understood that the density of correlations is already high, the lack of similar graphs, or mention of data, for normal weight individuals is of concern, especially if the intention is to show that obesity is linked to the expression of these microRNAs and that they, in part, explain the differences in expression of genes related to cortisol metabolism. Further if normal weight participants show similar trends, it lends weight to the hypothesis that the microRNAs are important in the expression of these genes. If miRNAs were not measured in this normal weight group, it is not clear from the methods that they were only used for non-miRNA studies.

Again, we thank the reviewer for this good point. Of course, miRNA concentrations and correlation studies were performed in all investigated tissues (Supplementary Figure S1). The failure to include results of mRNA/miRNA correlations for tissues from normal-weight subjects was clearly an oversight on our part. In the revised version of the manuscript, we have included these results as Supplementary Figure S2 and we have added a relevant comment in the Results section.

“Interestingly, a negative correlation between mRNA for HSDB112 and hsa-miR-579-3p and -5p was also observed in SAT-N tissues (Supplementary Figure S2).” (Page 7, lines 212-213)

9) Regardless of significance, data for adiponectin protein should be shown in figure 9 (line 222).

As suggested by the reviewer, we have added a graph on the correlation between cortisol and adiponectin concentrations to Figure 9.

“We observed a significant correlation between the cortisol concentration in SAT-O and mRNA levels for the gene encoding adiponectin (ADIPOQ), Figure 9a. SAT-O cortisol levels were also negatively correlated with adiponectin protein levels, but this correlation was not statistically significant (rs = -0.336, p > 0.05, Figure 9b).” (Page 8, lines 227-231)

Figure 9. Correlations of ADIPOQ mRNA (a) and adiponectin (b) concentrations with cortisol levels in subcutaneous adipose tissue of obese study participants (SAT-O). Black dots represent particular participants. Lines represent linear regression analysis with 95% CI interval. Adiponectin protein concentrations (ug) are normalized to the tissue protein content (mg).

Discussion:

While comprehensive, aspects of the discussion feel somewhat overstated in terms of impact- see specific comments below.

10) 252-258- It is unclear whether this statement is referring to a potential therapeutic approach of decreasing GR or whether it’s an intrinsic defense mechanism to limit metabolic complications. If the latter, the study does not support this. Metabolic improvement in adipocyte specific GR knockouts is not a validation that suppression of the GR is a defense mechanism in the course of obesity. Please clarify.

We do agree that the example of adipocyte-specific GR knockout animals is not a good one to support the hypothesis that the obesity-related decrease in expression of this receptor is a protective mechanism against the exacerbation of cortisol-induced obesity-related complications. Therefore, in the revised version of the manuscript, we have modified this section of the discussion as follows, and added the relevant reference.

“Since in the presented study, there were no difference in the concentration of this glucocorticoid in the tissues obtained from the obese and normal-weight subjects, one can conclude that the decreased expression of GR in the course of obesity can act as a protective mechanism preventing the dysfunctional adipose tissue from the additive, detrimental impact of glucocorticoids, aggravating e.g. insulin resistance or adipogenesis [24,25]. This is only speculation, but such a concept is plausible because studies in animal models show that high concentrations of glucocorticoids have a detrimental effect on adipogenesis and adipocyte function [24-26].” (Page 9, lines 264-271)

[26] Vali, A.; Dalle, H.; Loubaresse, A.; Gilleron, J.; Havis, E.; Garcia, M.; Beaupère, C.; Denis, C.; Roblot, N.; Poussin, K.; Ledent, T.; Bouillet, B.; Cormont, M.; Tanti, J. F.; Capeau, J.; Vatier, C.; Fève, B.; Grosfeld, A.; Moldes, M. Adipocyte Glucocorticoid Receptor Activation With High Glucocorticoid Doses Impairs Healthy Adipose Tissue Expansion by Repressing Angiogenesis. Diabetes 2024, 73, 211–224

11) 267-268- The authors state that differences in sex distribution may explain difference between their study and the Michailidou/Veilleux studies. The authors have the ability to do a sex stratified analysis, or analysis without males, especially given their significant proportions of females in their population; to see if this rectifies the difference, and should be included if they are claiming this as an explanation.

We thank the reviewer for this valid comment. Since we did not find any significant differences in the expression of the mRNAs and miRNAs investigated between the female and male subgroups, further analyses were performed for both sexes together. This led to the following change in the discussion.

“When considering the possible reasons for this discrepancy, the different characteristics of the study participants can be pointed out: whereas the tissues examined in our study were obtained from patients with grade III obesity undergoing bariatric surgery (mean BMI = 46.27 kg/m2), previous reports included participants with grade I obesity or overweight (mean BMI 32.7 kg/m2 and 27.1 kg/m2, respectively) [22,23].” (Page 9, lines 278-282)

We have also highlighted this in the results section.

“As there were no differences in expression between females and males for any of the investigated genes, all analyses described were performed together on all tissues without stratification by sex.” (Page 3, lines 123-125)

12) 273-4- A reference to figure 2 would be helpful here.

As suggested, we have added a reference to Figure 2 at this point in the discussion.

“Nevertheless, we observed a similar to previous researchers impact of obesity on the HSD11B1 expression between the visceral and subcutaneous depots: it was significantly higher in VAT of the normal-weight subject, while in the course of obesity its mRNA level were higher in SAT (Figure 2) [28].” (Page 9, lines 285-288)

13) 276-290- They have data refuting these conjectures of relative activity, given the lack of difference of active cortisol among all tissues (figure 8), recommend removing this paragraph or including reference to this data with a recontextualization of the differences.

We have removed this paragraph from the discussion, as the reviewer rightly suggested.

14) 297-298- The observations reported in this study are not consistent with this description. There is no correlation shown between the levels of 11BHSD1, H6PDH and concentrations of Cortisol (the only potential measure of 11BHSD1 activity reported in this study)

As rightly suggested by the reviewer, we have amended this paragraph in the discussion to reflect the results obtained.

“Previous studies have shown that the expression of these two genes and the activity of the enzymes they encode are positively correlated [29]. In our study, the expression profile of these two genes was very similar in the tissues examined.” (Page 10, lines 295-297)

15) 308-311- The associative data presented in this study cannot provide direction of these effects, it is possible that disruptions in Glucocorticoid activity could lead to changes consistent with type 2 diabetes. Consider language that conveys this uncertainty of direction.

&

16) 311-313 It is unclear what is meant by this sentence. Is it hat this previous review implied that metabolic perturbations in type 2 diabetes may lead to disruptions in GC action? Please clarify.

We thank the reviewer for this valid comment - in the revised version of the manuscript we have made an effort to avoid this kind of over-interpretation.

“Assessing whether lower expression of genes related to cortisol action and metabolism in adipose tissues of patients with type 2 diabetes may have clinical relevance requires verification by functional studies. Past studies suggested that response to glucocorticoids in type 2 diabetes can manifest as resistance or hyperactivity in a tissue-specific manner [reviewed in 30].” (Page 10, lines 306-310)

Reviewer 2 Report

Comments and Suggestions for Authors

The manuscript by Podraza et al is very interesting and innovative. The main aim of this study was to investigate the impact of obesity and weight loss on the expression of genes linked to cortisol metabolism and action in human adipose tissue and to verify if these changes may impact local cortisol and adipokines concentrations. Moreover, they investigated the important role of microRNA in the regulation of cortisol metabolism in the adipose tissue in the course of obesity. However, I have some comments to raise with the authors to improve the quality of the work.

1-    The aim of the work needs to be made clearer and more incisive in both the introduction and discussion sections;

2-    The authors must emphasize in the introduction section the inflammatory status of subcutaneous and visceral white adipose tissue during obesity (https://pubmed.ncbi.nlm.nih.gov/32158768/; https://pubmed.ncbi.nlm.nih.gov/37111329/);

3-    The study was conducted on 75 individuals with obesity (62 females and 13 males). In this regard, the work has one significant limitation, which is the incorrect ratio between the number of male and female patients recruited in the study. It must be considered that female sex hormone levels can strongly influence both cortisol levels but also the gene expression of the genes and miRNAs studied. I would ask the authors to emphasize this considerable limitation of the study in the discussion section.

Comments on the Quality of English Language

Minor editing of English language

Author Response

Reviewer 2

The manuscript by Podraza et al is very interesting and innovative. The main aim of this study was to investigate the impact of obesity and weight loss on the expression of genes linked to cortisol metabolism and action in human adipose tissue and to verify if these changes may impact local cortisol and adipokines concentrations. Moreover, they investigated the important role of microRNA in the regulation of cortisol metabolism in the adipose tissue in the course of obesity. However, I have some comments to raise with the authors to improve the quality of the work.

We would like to thank the reviewer for the positive reception of our work and the comments, which will undoubtedly improve the quality of the manuscript.

1) The aim of the work needs to be made clearer and more incisive in both the introduction and discussion sections

We thank the Reviewer for this valid comment. In the revised version of the manuscript, we have added relevant passages in the introduction and discussion to emphasize the aim of this work

“Given the phenotypic similarity between obesity and hypercortisolaemia, it is plausible that excessive body weight will affect cortisol metabolism in adipose tissue. Therefore, the main aim of this study was to investigate the impact of obesity and weight loss on the expression of genes key to cortisol metabolism and action (Figure 1) in human adipose tissue. We also aimed to verify whether these changes could affect local concentrations of cortisol and selected adipokines. Then, given the role of cortisol in the regulation of carbohydrate metabolism and insulin sensitivity, we also investigated how the diagnosis of diabetes affected the expression of the of the genes studied. Finally, in the search for potential factors responsible for obesity-related changes in the expression of genes related to cortisol action and metabolism in adipose tissues, we investigated whether microRNA interference could play such a role.” (Page 3, lines 108-118)

“The aim of the present study was to investigate the effects of obesity on cortisol levels and the expression of genes key to its synthesis and function in adipose tissue. Given the role of cortisol in the regulation of glycaemia and insulin sensitivity, we also investigated how the expression of these genes changes in the tissues of diabetic patients. Another aim of the study was to assess whether adipose tissue cortisol concentrations correlate with local inflammation severity and impact the expression of selected adipokines. Finally, attempts were also made to correlate the results of the gene expression analysis with the levels of certain microRNAs to see whether this epigenetic mechanism might be responsible for obesity-related changes in glucocorticoid metabolism and action.” (Page 9, lines 241-250)

2)  The authors must emphasize in the introduction section the inflammatory status of subcutaneous and visceral white adipose tissue during obesity (https://pubmed.ncbi.nlm.nih.gov/32158768/; https://pubmed.ncbi.nlm.nih.gov/37111329/);

Following the Reviewer's valid comment in the introduction, we have added a relevant section on ongoing inflammation in adipose tissue, referring to the relevant references.

“An additional consideration for studying obesity-related changes in adipose tissue cortisol metabolism is that glucocorticoids are key regulators of the inflammatory response. Obesity-related dysfunction of adipose tissue is manifested by infiltration of inflammatory cells and a change in secreted adipokines towards those with a pro-inflammatory profile. Inflammatory mediators exacerbate insulin resistance in adipose tissue through auto- and paracrine effects and, when released into the circulation, interfere in an endocrine manner with the function of other organs and tissues, contributing to the development of obesity-related complications [15,16].” (Page 3, lines 93-101)

[15] Chait, A.; den Hartigh, L.J. Adipose Tissue Distribution, Inflammation and Its Metabolic Consequences, Including Diabetes and Cardiovascular Disease. Front Cardiovasc Med 2020, 7, 22.

[16] Petito, G.; Cioffi, F.; Magnacca, N.; de Lange, P.; Senese, R.; Lanni, A. Adipose Tissue Remodeling in Obesity: An Overview of the Actions of Thyroid Hormones and Their Derivatives. Pharmaceuticals 2023, 16, 572. Pharmaceuticals 2023, 16, 572.

3) The study was conducted on 75 individuals with obesity (62 females and 13 males). In this regard, the work has one significant limitation, which is the incorrect ratio between the number of male and female patients recruited in the study. It must be considered that female sex hormone levels can strongly influence both cortisol levels but also the gene expression of the genes and miRNAs studied. I would ask the authors to emphasize this considerable limitation of the study in the discussion section.

We agree with the reviewer that the disproportion in the number of female and male participants is a limitation of our study, although this is a typical arrangement for patients undergoing bariatric surgery. Since, as we noted in the Results section, we found no significant differences in the expression profile of the genes studied between the sexes, all analyses were conducted jointly (for females and males). We have included a comment on this in the discussion section on limitations of the study.

We agree with the reviewer that the disproportion in the number of female and male participants is a limitation of our study, although this is a typical arrangement for patients undergoing bariatric surgery. Since, we found no significant differences in the expression profile of the genes studied between the sexes (Supplementary Table S1), all analyses were conducted jointly (for females and males). We have included a comment on this in the Results and Discussion sections on limitations of the study.

“As there were no differences in expression between females and males for any of the investigated genes, all analyses described were performed together on all tissues without stratification by sex.” (Page 3, lines 123-125)

“Another potential limitation of our work is that the studied cohort was predominantly female (which is typical for patients undergoing bariatric surgery) and local differences in sex hormone levels could affect the results. However, in our cohort, we found no significant differences in the expression profile of the genes studied between the sexes.” (Page 11, lines 360-364)

Round 2

Reviewer 1 Report

Comments and Suggestions for Authors

I have no further comments. Thank you for taking the time to address my recommendations in a comprehensive fashion.

Reviewer 2 Report

Comments and Suggestions for Authors

The authors fully answered the questions raised to them. The quality of the work has improved considerably, and I consider the work suitable for publication in IJMS journal

Comments on the Quality of English Language

Minor editing of English language